# Research Progress on Micro (Nano)Plastics Exposure-Induced miRNA-Mediated Biotoxicity

**DOI:** 10.3390/toxics12070475

**Published:** 2024-06-29

**Authors:** Ting Chen, Qizhuan Lin, Changyong Gong, Haiyang Zhao, Renyi Peng

**Affiliations:** Institute of Life Sciences & Biomedicine Collaborative Innovation Center of Zhejiang Province, College of Life and Environmental Science, Wenzhou University, Wenzhou 325035, China; 23461044002@stu.wzu.edu.cn (T.C.); 23461044010@stu.wzu.edu.cn (Q.L.); 23461044004@stu.wzu.edu.cn (C.G.)

**Keywords:** Micro- and nano-plastics, miRNA, lncRNA, circRNA, biotoxicity, disease

## Abstract

Micro- and nano-plastics (MNPs) are ubiquitously distributed in the environment, infiltrate organisms through multiple pathways, and accumulate, thus posing potential threats to human health. MNP exposure elicits changes in microRNAs (miRNAs), long noncoding RNAs (lncRNAs), and circular RNAs (circRNAs), thereby precipitating immune, neurological, and other toxic effects. The investigation of MNP exposure and its effect on miRNA expression has garnered increasing attention. Following MNP exposure, circRNAs serve as miRNA sponges by modulating gene expression, while lncRNAs function as competing endogenous RNAs (ceRNAs) by fine-tuning target gene expression and consequently impacting protein translation and physiological processes in cells. Dysregulated miRNA expression mediates mitochondrial dysfunction, inflammation, and oxidative stress, thereby increasing the risk of neurodegenerative diseases, cardiovascular diseases, and cancer. This tract, blood, urine, feces, placenta, and review delves into the biotoxicity arising from dysregulated miRNA expression due to MNP exposure and addresses the challenges encountered in this field. This study provides novel insights into the connections between MNPs and disease risk.

## 1. Introduction

Micro- and nano-plastics (MNPs), which are plastic fragments or particles measuring less than 5 mm or 1 μm, respectively, are widely distributed across various ecosystems. They originate from the degradation of plastic products, the abrasion of automobile tires, and the use of additives in daily products; MNPs can infiltrate biological organisms through contact or ingestion [1]. In Europe, it is conservatively estimated that 1270 to 2130 tons of MNPs per million inhabitants are released into the urban environment, while 63,000 to 430,000 tons of MNPs are released into cropland. In North America, approximately 44,000 to 300,000 tons of MNPs are released into fields each year. In China, existing research indicates that the abundance of MNPs in agricultural soil ranges from 4.94 items/kg downstream of the Yangtze River to 40,800 items/kg in Yunnan Province [2,3]. Research indicates that prolonged exposure to MNPs may pose various health risks. Exposure to MNPs across diverse organisms or cell types has demonstrated that MNPs can induce various forms of tissue, cellular, or molecular toxicity. These effects encompass developmental, reproductive, cardiovascular, neurological, and other physiological systems. For instance, studies in adult zebrafish have revealed that MNPs can penetrate the brain and inhibit acetylcholinesterase activity, thereby altering neurotransmitter levels and inducing neurotoxicity. Furthermore, MNPs can infiltrate cells, perturb intracellular signaling pathways, and disturb immune homeostasis, ultimately triggering immunotoxicity in marine organisms and mammals [4,5]. Li et al. found that the Nrf2/NF-κB, Bcl-2/Bax, and AKT signaling pathways were activated in the chicken thymus after exposure to polyethylene microplastics (PS-MPs), triggering thymic inflammation, apoptosis, and autophagy, thereby inducing immunotoxicity. Xuan et al. discovered in RAW264.7 cells that NPs could induce apoptosis, cause cell cycle arrest, and trigger dysregulated innate immunity and inflammatory responses through the cGAS/STING pathway [6,7]. Therefore, it is important to select appropriate techniques to characterize MNPs in cells, such as high-throughput single-molecule tools and human biomonitoring (HBM). Giuseppina et al. summarize in their paper that HBM is capable of detecting MNPs in the upper and lower respiratory tracts, blood, urine, feces, placenta, and breast milk, and also provide a related risk assessment of human exposure to MPs [8,9]. Notably, prolonged exposure to MNPs may significantly increase the risk of cancer, cardiovascular diseases, and neurodegenerative diseases. This increased risk is closely associated with mitochondrial dysfunction, inflammatory responses, and aberrant apoptosis. Studies have found that PS-NPs promote mitochondrial oxidative stress, activate the P38/Erk MAPK signaling pathway, and block the autophagy pathway. These effects lead to structural changes in mitochondria and a reduction in membrane potential. Liang et al. revealed that PS-NPs-induced Parkinson’s disease (PD)-like neurodegeneration in mice was closely related to mitochondrial dysfunction [10,11,12,13].These adverse effects are attributed to the dysregulation of gene and protein expression induced by MNP exposure via a process intricately linked to miRNA dysregulation.

MicroRNAs (miRNAs) are small endogenous RNA molecules that post transcriptionally regulate gene expression. miRNAs are produced from noncoding regions of the genome and transcribed into primary miRNAs with the help of RNA polymerase II. miRNAs are approximately 19 to 25 nucleotides in length and are widely present in various eukaryotes. miRNAs can influence the expression and translation of mRNA, thereby impacting biological processes such as growth and development, cell differentiation, embryogenesis, organogenesis, and apoptosis [14]. miRNAs can also bind to circular RNAs (circRNAs), acting as sponges and regulating the expression of downstream proteins through various biological activities and mechanisms. Long noncoding RNAs (lncRNAs) are a class of transcripts exceeding 200 nucleotides in length. Although they do not encode proteins, lncRNAs participate in and regulate various biological processes, including DNA methylation, histone modification, posttranscriptional RNA regulation, and the regulation of protein translation. lncRNAs are closely associated with different physiological and pathological processes. As a novel environmental pollutant, microplastics can modulate lncRNA levels and regulate the expression of target genes by acting as competing endogenous RNAs (ceRNAs) that bind to miRNAs [15,16]. This interaction leads to changes in biological effects, ultimately causing a cascade of toxicity, injury, and disease.

This review summarizes the alterations in miRNA levels associated with bodily damage due to MNP exposure, emphasizing the relationship between miRNA dysregulation and related diseases. This study provides valuable insights for further exploration in this field and suggests future research directions.

## 2. miRNAs Are Involved in Neurotoxicity and Immunotoxicity Caused by Microplastic Exposure

Exposure to MNPs can induce neurotoxicity mediated by miRNAs, which is characterized by abnormal brain activity in response to harmful substances. MNPs accumulate in the brain, leading to neurotoxicity, which is closely linked to alterations in miRNA levels. miRNAs exhibit tight temporal and spatial regulation within the nervous system. When external conditions change, the expression profile of miRNAs in nerve cells will change, making the central nervous system a vulnerable target of MNPs. RNA sequencing has revealed that 29 miRNAs in synapses are differentially expressed when exposed to MNPs, resulting in synaptic dysfunction [17,18,19]. Previous studies have demonstrated that PS-MPs of varying sizes can accumulate in mouse brains, thereby disrupting the blood-brain barrier and causing neurotoxicity, resulting in learning and memory dysfunction. Additionally, exposure to PS-NPs alters the expression of circRNAs and miRNAs, thus regulating the expression of mRNAs associated with synaptic function in the prefrontal cortex (PFC) of mice. Among the affected miRNAs, *miR-152-3p* and *let-7g-5p* play crucial roles. Abnormal activation of *miR-152-3p* targets DNMT1 and contributes to defective neuronal differentiation and rapid cell death [19,20]. Moreover, overexpression of *miR-199a-5p* induces apoptosis and inflammation in neural cells, thereby promoting the development of depression through WNT2 modulation of the CREB/BDNF signaling pathway (Figure 1). Furthermore, differential expression of 67 circRNAs has been reported, and *circ_Arhgap32*, a host gene, was closely associated with cognitive decline [21,22,23,24].

Immunotoxic effects are mediated by immune regulation. miRNA serves as a biomarker for the regulation of immune response gene expression, and changes in its levels can specifically indicate immunotoxicity caused by exogenous biological or nonbiological factors [25]. When MNPs enter the systemic circulation, immune cells engulf microplastics, resulting in damage. Shi et al. found that 65 of 196 miRNAs were up-regulated and 131 were down-regulated in mice treated with MPS, playing an important role in immunity, inflammatory response, and cell interaction. Sun et al. reported that blood cockles exhibited significant immunotoxicity when exposed to MPs. Kwon et al. reported that the process involving PS-MPs altered the expression of gene clusters related to the immune response, immunoglobulins, and several related microRNAs. Microarray analysis confirmed that seven microRNAs were downregulated and two microRNAs were upregulated, and these changes posed potential risks for microglial immune activation. Finally, Diao et al. found that exposure to PS-MPs led to a decrease in the expression of *miR-25-5p* in lymphocytes. The *miR-25-5p*/MCU axis plays a key role in immune damage [26,27,28,29].

## 3. Mechanisms of miRNAs in Microplastic-Induced Injury

### 3.1. Role of miRNAs in MNP-Induced Injury

MNP exposure can modulate miRNA expression levels, leading to injury. *Caenorhabditis elegans* exhibits heightened sensitivity to various environmental pollutants, with exposure to PS-NPs resulting in the upregulation of *miR-35* and *miR-354* expression and the downregulation of *miR-39*, *miR-76*, *miR-794*, and *miR-1830* expression, thereby influencing environmental resistance [30,31,32]. miRNAs play a crucial role in mediating the toxic responses induced by MNP exposure. Hua et al. found that *miR-38* expression decreased when exposed to 20 nm PS-NPs. *miR-38* can bind to its 3′ UTR to inhibit NDK-1 activity. NDK-1 is one of the important downstream direct targets of *miR-38* in controlling the toxicity of PS-NPs across generations. In *Caenorhabditis elegans*, NDK-1 plays a role upstream of the KSR-1/2 signaling cascade in controlling intergenerational toxicity. Therefore, in *Caenorhabditis elegans*, reduced *miR-38* inhibits the NDK-1-KSR-1/2 signaling cascade, mediating intergenerational PS-NP-induced toxicity. Interestingly, upregulation of *miR-38* expression mitigated transgenerational toxicity. These findings suggest that miRNAs and their downstream targets play crucial roles in regulating the toxic response to PS-NPs. Huang et al. demonstrated that both 50 nm PS-NPs and 5 μm PS-MPs reduced *miR-126a-3p* expression, along with inducing oxidative damage to intestinal cells via the PI3K-Akt pathway, promoting apoptosis, disrupting the intestinal flora, and impairing intestinal barrier function. Li et al. discovered that mice exposed to tire wear microplastic particles (TWMPs) via inhalation experienced pulmonary toxicity, which attenuated *miR-1a-3p* expression. This attenuation led to increased secretion of twinfilin-1, inhibited F-actin formation, triggered cytoskeletal rearrangement, and exacerbated pulmonary fibrosis [33,34,35]. Moreover, miRNA expression is closely correlated with reactive oxygen species (ROS) generation. Studies have shown that *miR-35* expression is upregulated in *Caenorhabditis elegans* exposed to PS-NPs, resulting in ROS overproduction in the gut [32,36]. This ROS overproduction induces oxidative stress and leads to mitochondrial DNA damage, ultimately resulting in mitochondrial dysfunction. Wang et al. found that PS-MPs increased the production of extracellular vesicles (EVs) in renal tubular cells, inducing the production of ROS and endoplasmic reticulum (ER) stress-related proteins in fibroblasts. The miRNAs contained within these EVs spread into the extracellular space, influencing the expression of oncogenes and accelerating the process of cancer [37,38]. These findings underscore the mechanism by which PS-NPs disrupt miRNA-mediated cascade reactions, leading to injury.

### 3.2. CircRNA Acts as a “Sponge” of miRNA and Plays a Role in Injury Induced by MNP Exposure

CircRNAs serve as miRNA sponges that exert regulatory control over diverse physiological processes. Chen et al. discovered that *circ_Trpm7* functions as a sponge for *miR-383-3p*, thereby modulating the expression of Hras, Nlgn1, Ntng1, Ppp3ca, and Rpl5 in mice exposed to PS-NPs, which play a pivotal role in cognition. Consequently, *circ_Trpm7* impacts synaptic differentiation and intracellular signaling. The researchers further predicted that *circRNA4186* may serve as a sponge for *miR-199b-5p*, thereby modulating the expression of Igsf9b, a crucial factor in synaptic development that could play a role in the toxicity of PS-NPs and cognitive dysfunction [19,39]. Following exposure to PS-MPs, an increase in the number of senescent cells was observed in rat lung tissue, concomitant with an increase in the level of *circ_kif26b*. *Circ_kif26b* functions as a sponge for *miR-346-3p* and facilitates the expression of the target gene P21. P21 is closely associated with inflammation, and its elimination can mitigate the inflammatory response, thus conferring protection against lung inflammation mediated by other proinflammatory stimuli [30,40]. Concurrently, the levels of the senescence-associated markers p16 and p27 were also elevated, thereby prompting the release of the senescence-associated secretory phenotype (SASP), triggering an inflammatory response, and expediting cellular senescence. Eif4e plays a crucial role in nervous system function. Previous studies have revealed that *circ_Phkb* can act as a sponge for *miR-216b-5p*, *miR-3572-3p*, and *PC-3p-25230_17* under MNP exposure, thereby modulating the expression of the Eif4e and Nrp2 genes. Nevertheless, disruptions in Eif4e translation are implicated in synaptic protein loss via the TrkB/BDNF signaling pathway, ultimately culminating in inflammation [26,41]. Moreover, the in vivo deposition of MNPs, which results in the differential expression of circRNAs, can directly induce the onset of inflammation (Figure 2a).

### 3.3. lncRNAs, as ceRNAs of miRNA, Are Involved in Injury Induced by MNP Exposure

miRNAs can recognize and bind to the RNA-induced silencing complex (RISC), which allows them to target lncRNAs and exert counterregulatory effects on lncRNA expression. As a highly abundant and evolutionarily conserved miRNA, *miR-21* is ubiquitously expressed across various cell types, and alterations in its expression are closely associated with numerous diseases. *miR-21* interacts with the lncRNA growth arrest-specific transcript 5 (GAS5) to regulate miRNA-mediated gene silencing [42,43]. Exposure to polyethylene microplastics (PE-MPs) reduces the level of *miR-21* in muscle tissue. *miR-21* can target and regulate GAS5, and its overexpression promotes apoptosis. *miR-21* is also negatively correlated with the *lncRNA NEAT1*. Increased levels of NEAT1 can promote the expression of MIOX by competitively binding to *miR-362-3p*, thereby increasing the production of ROS, reducing intracellular NADPH and GSH levels, and inducing ferroptosis [43,44,45,46]. Additionally, *lncRNA-H19* is a key pathogenic factor in ischemia-reperfusion-induced inflammation that forms a competing endogenous RNA network (ceRNET) with *miR-21* in the ischemic cascade. Elevated levels of *lncRNA-H19* lead to an imbalance in the NLRP3/6 inflammasome, thereby inducing microglial pyroptosis, cytokine overproduction, and neuronal death (Figure 2b). Huang et al. found that MEG3 and LRP6 can bind with *miR-21,* competitively inhibiting the mTOR pathway and inducing lipid accumulation in cells. Additionally, MEG3, acting as a ceRNA for *miR-21*, can enhance the sensitivity of cervical cancer cells to cisplatin [47,48]. However, there are few studies on lncRNAs acting as ceRNAs for other related miRNAs under MNP exposure, indicating that this idea could be a potential focus for future research.

## 4. miRNA-Mediated Mechanism of MNP Toxicity

With respect to MNP exposure, relevant literature on specific cellular or molecular toxicity responses has focused mainly on miRNAs, while studies on lncRNAs and circRNAs are less common. Nevertheless, lncRNAs and circRNAs are indispensable for the regulation of physiological processes in the body. Therefore, these two ncRNAs should be emphasized in future research on the biotoxicity of MNPs. Based on the results of existing studies, we have summarized the miRNA-mediated toxicity mechanisms of various types of MNPs.

MiRNAs play crucial regulatory roles in the development of inflammation. Exposure to PE-MPs significantly reduces the expression of *miR-132*, an important transcriptional regulator of several inflammation-related mediators. *miR-21*, which exerts anti-inflammatory effects in various inflammation-related diseases in humans and mice, can inhibit inflammation by targeting IRAK4, thereby negatively regulating cytokine production in the NF-κB pathway and preventing excessive inflammation [49,50,51]. Liu et al. reported that PE-MPs reduce *miR-21* levels, increase IRAK4 expression, and activate the NF-κB pathway. In mouse models, inhibition of *miR-21* resulted in increased levels of TNF-α, IL-17, and IL-21, thereby exacerbating inflammatory responses. Renal fibrosis is the final common stage of all chronic kidney diseases, characterized by the infiltration of inflammatory cells into the damaged kidney. Studies have shown that *miR-21* plays a dynamic role in the inflammatory response and fibrosis, exerting a long-term impact on chronic inflammation [44,52,53]. Abdelrahman et al. demonstrated that exposure to zinc oxide nanoparticles significantly increased the serum levels of *miR-21-5p*, *miR-122-5p*, *miR-125b-5p*, and *miR-155* in mice. *miR-21-5p*, a mature miRNA at the 5′ end of *miR-21*, helps maintain T-cell-derived skin inflammation. *miR-155* expression is upregulated in synovial macrophages and synovial fluid macrophages of patients with rheumatoid arthritis (RA), leading to increased production of proinflammatory cytokines. Additionally, *miR-132* and *miR-21* target IL-1β, which can polarize macrophages toward an anti-inflammatory phenotype and induce the expression of the anti-inflammatory cytokine IL-10 [54,55,56,57]. After exposure to PS-MPs, *miR-346-3p* expression in rat alveolar epithelial cells was upregulated, thereby exerting an inflammatory effect. However, using a *miR-346-3p* inhibitor reversed this inflammatory effect [30]. Thus, MNPs can modulate miRNA activity in vivo and subsequently induce an inflammatory response.

The level of miRNA is closely related to MNP-induced apoptosis. *miR-132* has the ability to inhibit apoptosis; upregulation of *miR-132* expression can promote the proliferation of pancreatic cancer cells and inhibit apoptosis. Zhu et al. reported that the expression of *miR-132* was significantly reduced in the presence of 8 μm PE-MPs, which diminished its inhibition of the target gene CAPN. CAPN is a key enzyme in the apoptosis pathway, and a significant increase in CAPN levels can lead to mitochondrial damage. This damage results in the release of Ca^2+^ from mitochondria into the cytoplasm, thereby inducing the production of proapoptotic factors such as AIF, Bax, and cytochrome c. These changes disrupt the normal function of the respiratory chain, trigger the production of a large amount of ROS, and ultimately induce apoptosis. *miR-1* also plays a key role in apoptosis. When mice are exposed to titanium dioxide nanoparticles, *miR-1* expression is significantly upregulated. The increased *miR-1* expression in cells can inhibit the proliferation of cardiomyocytes and promote apoptosis [49,58,59]. Finally, *miR-34a* plays a crucial role in apoptosis. Studies have found that overexpression of *miR-34a* can induce ROS production and trigger G0/G1 cell cycle arrest, ultimately leading to apoptosis. Therefore, changes in *miR-34a* levels offer a direction for subsequent studies on MNPs-induced stress (Table 1) [60,61].

## 5. Changes in miRNAs Induced by MNPs Are Involved in the Occurrence and Development of Related Diseases

MiRNAs have garnered considerable attention in research on MNP exposure and potential disease risks. Alterations in miRNA levels play pivotal roles in the pathogenesis of various diseases, including cancer, neurodegenerative diseases, cardiovascular diseases, and immune diseases. MNPs have the capacity to alter the levels of miRNA, thereby potentially increasing the risk of disease.

In the presence of MNPs, the level of miRNA is altered, which significantly impacts the occurrence and progression of cancer. Exposure to PS-MPs or cadmium decreases the level of *miR-199a-5p*. As a tumor suppressor gene, increased levels of *miR-199a-5p* significantly inhibit tumor growth and metastasis [62,63]. Additionally, miRNA levels play a critical role in PD. *miRNA-124* demonstrates pro-neurogenic and neuroprotective effects, with miR-124-3p-loaded small extracellular vesicles inducing neuronal differentiation in vitro in subventricular zone neural stem cell cultures. Moreover, *miRNA-124* protects N27 dopaminergic cells from 6-hydroxydopamine-induced toxicity, suggesting its potential as a therapeutic candidate for Parkinson’s disease [64]. miRNAs also play a crucial role in the progression of cardiovascular diseases. Studies have shown that exposure to MNPs in carp leads to the downregulation of *miR-25-5p* expression. Furthermore, in patients with hypertension, *miR-25-5p* is closely associated with the risk of heart failure [29]. *miR-122* is enriched and expressed in the liver, playing a central role in the development, differentiation, homeostasis, and function of the liver. Long et al. found that inhibition of miR-122 can protect hepatocytes from the effects of lipid metabolism disorders such as non-alcoholic fatty liver disease. It also inhibits adipogenesis by increasing Sirt1 and activating the AMPK pathway. Chen et al. reached a similar conclusion, finding that high levels of *miR-122* can promote adipogenesis, cause liver cell damage, liver injury, and elevated serum lipid levels. However, there are currently no reports on whether exposure to MNPs increases *miR-122* levels. This could be a focus for future research [65,66,67].

MiRNAs play a significant role in autoimmunity. Among them, *miR-21* is crucial not only in the development of cardiovascular disease and cancer but also in regulating immune diseases. The balance between Th17 and Treg cells is implicated in numerous autoimmune diseases, such as inflammatory bowel disease (IBD), rheumatoid arthritis (RA), and multiple sclerosis (MS), and *miR-21* regulates their homeostasis. Under exposure to PS-NPs, the level of *miR-21* is altered, leading to increased expression in Th17 cells. Mice lacking *miR-21* exhibit defects in Th17 differentiation and resistance to experimental autoimmune encephalomyelitis, which indicates autoimmune defects [68,69]. Anti-inflammatory polarized macrophages can alleviate systemic lupus erythematosus (SLE), and exosome-mediated delivery of *miR-21* plays a crucial role in macrophage polarization. Additionally, lupus nephritis, a serious complication of SLE, involves the overexpression of *miR-21* in peripheral blood monocytes, which contributes to SLE pathophysiology. According to existing research reports, the pathogenesis of SLE involves innate immunity. *miR-21* can trigger the innate immune response by binding to Toll-like receptors (TLR) along with ssRNA viruses. This interaction stimulates innate immunity and affects gene expression through epigenetic modifications. Additionally, *miR-21* supports abnormal cytokine release, cell subset differentiation, overactive B cells, and autoantibody production, thereby contributing to the pathogenesis of SLE [70,71,72,73]. Following exposure to PS-NPs, *miR-124* exhibited significantly different expression. Previous studies have indicated that ANTXR2 is one of the risk loci associated with ankylosing spondylitis (AS). Overexpression of *miR-124* can inhibit ANTXR2 expression and induce autophagy, thereby participating in the pathological process of AS [74]. Therefore, miRNAs play a significant role in the pathophysiological processes of immune diseases.

## 6. Conclusions

MNP pollution has become an environmental problem that urgently needs to be addressed. Exposure to MNPs can cause changes in the expression levels of key miRNAs in organisms, thereby inducing neurotoxicity and immunotoxicity and increasing the risk of disease. MNPs can also indirectly alter miRNA expression through lncRNAs; lncRNAs act as ceRNAs of miRNAs and impact the transcription and expression of downstream target genes. Additionally, circRNAs can sponge miRNAs, thereby affecting normal physiological regulatory processes. The changes in miRNA induced by exposure to MNPs are involved in mediating inflammation, oxidative stress, mitochondrial dysfunction, and programmed cell death. These alterations may lead to neurodegenerative diseases, immune diseases, and cancer.

To date, data on the involvement of miRNAs in MNP-mediated biotoxicity are limited, and the differential expression of lncRNAs and circRNAs in MNP-induced adverse reactions is even less well studied. Therefore, future studies should focus on elucidating the molecular mechanisms triggered by these key ncRNAs. Notably, crosstalk between different targets via the ceRNA network and sponge interactions may be an important mechanism of MNP-induced injury. This aspect could be the focus of further studies to elucidate the potential mechanism of MNP toxicity. As a result, changes in the level of ncRNA can help us further understand the link between the MNPs and pathophysiology.

## Figures and Tables

**Figure 1 toxics-12-00475-f001:**
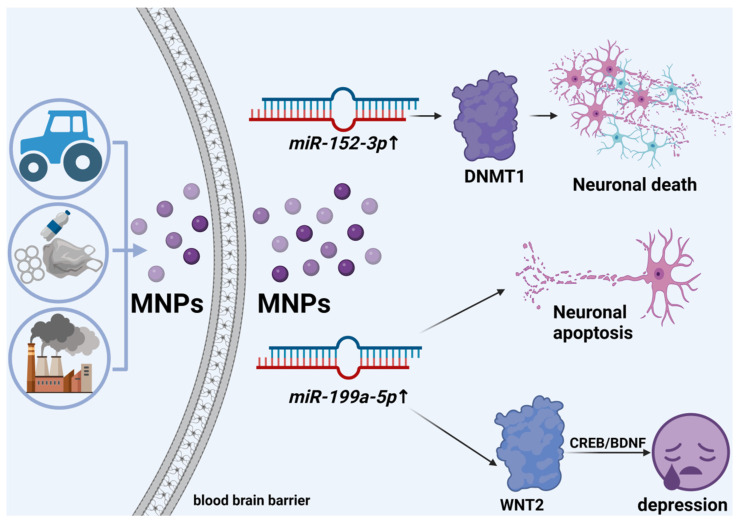
Diagram of the mechanism of action by which MNPs cause neurotoxicity. MNPs, which are derived from the degradation of plastic products, automobile tire wear, and industrial product degradation in the environment, can penetrate the blood-brain barrier. This penetration induces increases in the levels of *miR-152-3p* and *miR-199a-5p*, which target DNMT1, leading to neuronal death and apoptosis. Elevated *miR-199a-5p* levels can also target WNT2 and promote the progression of depression through the CREB/BDNF signaling pathway.

**Figure 2 toxics-12-00475-f002:**
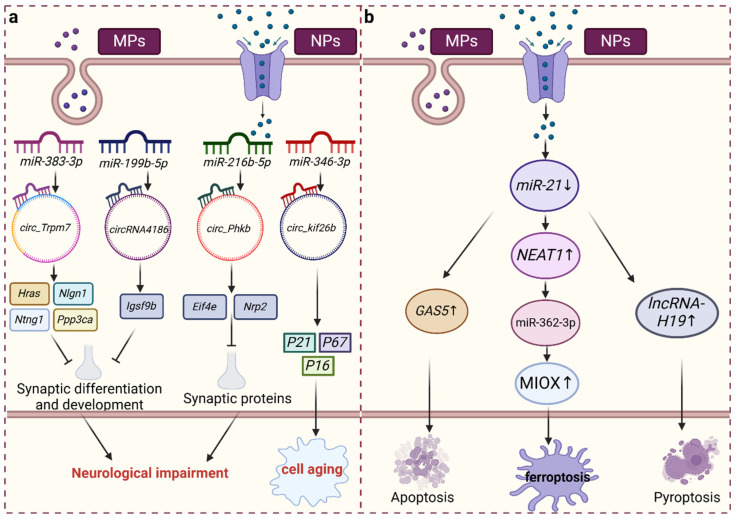
CircRNA-sponging miRNA and ceRNA regulatory networks under the action of MNPs. Under MNP exposure, circRNAs are differentially expressed and act as sponges for miRNAs, thereby affecting the expression of downstream genes. This results in neurological impairment and cellular senescence (**a**). Under MNP exposure, decreased levels of *miR-21* allow lncRNAs to act as ceRNAs and induce programmed cell death (**b**).

**Table 1 toxics-12-00475-t001:** Effects of MNPs on miRNA.

MNPs Type and Particle Size	Species/Cell	Changes in miRNA	Phenotype	References
PS-MPs (0.1/1 μm)	mice	*miR-199a-5p* ↓ *miR-106a-5p* ↓ *miR-101a-3p* ↑	Injury to reproduction	[62]
PS-MPs (100 nm)	MLE12 cells	*miR-346-3p* ↓	cell senescence	[30]
TWMP (100 nm)	BEAS-2B cells	*miR-1a-3p* ↓ *miR-206-3p* ↓ *miR-381-3p* ↓ *miR-204-5p* ↓	Rearrangement of the cytoskeleton	[35]
PE-PMs (8 μm)	Carp	*miR-132* ↓	apoptosis	[49]
PS-NPs (50 nm/5 μm)	rat	*miR-126a-3p* ↓ *miR-27a-5p* ↓	inflammation	[34]
PE-MPs (8 μm)	Carp	*miR-21* ↓ *miR-203* ↓ *miR-181* ↑	oxidative stress	[44]
PM (250 μM)	Carp	*miR-25-5p* ↓	mitochondrial dysfunction	[29]
PS-NPs (20 nm)	*Caenorhabditis elegans*	*mir-38* ↓	transgenerational toxicity	[33]
PS-NPs (100 nm)	*Caenorhabditis elegans*	*mir-39* ↓ *mir-76* ↓ *mir-794* ↓ *mir-1830* ↓	oxidative stress	[32]
		*miR-35* ↑ *miR-38* ↑ *miR-354* ↑		

## Data Availability

The data underlying this article are available in the corresponding references.

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
