# Peer review of "Research Progress on Micro (Nano)Plastics Exposure-Induced miRNA-Mediated Biotoxicity"

_toxics, 2024, doi:10.3390/toxics12070475_

Round 1
Reviewer 1 Report
Comments and Suggestions for Authors
I have no issues to report or relevant suggestions to improve the quality of present manuscript.
However, among all reported studies, I can suggest to consider also some recent findings on impact of MNPs on extracellular vesicles homeostasis, that are known to consequently affect also miRNAs.
Moreover I just recommend minor editorial check of English language.
Reviewer 2 Report
Comments and Suggestions for Authors
The manuscript titled “Research Progress on Micro( Nano)Plastics Exposure-induced miRNA-Mediated Biotoxicity” by Chen, T. et al. is a Review work where the authors outlined the most recent advances in the impact of micro- and nano-plastic (MNP) toxicity in living cells and how they can cause RNA dysregulation and the ocurrence of many human diseases. The most relevant bullet points highlighted by the authors could open new gates in the design of the next-generation of therapies exploiting the molecular targets affected by the MNP exposition which have been depicted in this Review work. The manuscript is generally well-written and this is a topic of growing interest.
However, it exists some points that need to be addressed (please, see them below detailed point-by-point) to improve the scientific quality of the submitted manuscript paper before this article will be consider for its publication in Toxics.
1) KEYWORDS. The authors should consider to also add the term “circRNAs” in the keyword list combined to the already existing “miRNA” and “lncRNA”.
2) INTRODUCTION. “Micro- and nano-plastics (….) distributed acros various ecosystems. They originate from the degradation of plastic products (…) in daily products” (lines 26-29). Could the authors provide quantitative data details concerning the worldwide micro- and nano-plastics generation? This will significantly aid the potential readers to better understand the significance of this Review work.
3) Then, it should be also advisable to mention suitable techniques to characterize the preence of micro- and nano-plastics in liquid cellular environments like biomonitoring [1], or single-molecule tools [2].
[1] https://doi.org/10.1016/j.envres.2023.116966
[2] https://doi.org/10.22034/IAR.2022.1965012.1317
4) “2. miRNAs are involved in neurotoxicity and immunotoxicitycaused by microplastic exposure” (lines 65-99). “Previous studies (…) PS-MPs (…) dysfunction” (lines 71-73). The full-name should be defined the first time that a term appears. Then, the abbreviation should be placed between brackets. This comment should be taken into account for the rest of the main manuscript body text.
5) “3. Mechanism of miRNA in microplastic-induced injury” (lines 100-174). This section is clearly explained. No actions are requested from the authors.
6) “4. mi-RNA mediated mechanism of MNP injury” (lines 175-217). Even if I agree with the information provided in this section it should not be neglected the role of miR-34a in the apoptosis regulation. miR-34a can induce cell cycle arrest making it a pivotal actor in the response to MNP-induced stress.
7) “5. Changes in miRNA induced by MNPs are involved in the occurrence and development of related diseases” (lines 218-254). The authors need to expand the discussion about liver diseases and how the alterations of miR-122 expression due to MNP exposure can lead to hepatitis or hepatocellular carcinoma clinical cases.
8) CONCLUSIONS. This section perfectly remarks the most relevant outcomes found by the authors in this field and about the future line actions to pursue this research. The authors should add a brief statement to discuss about the promising open perspectives.
Comments on the Quality of English LanguageThe manuscript is generally well-written albeit it may be desirable if the authors could recheck it in order to polish those final details susceptible to be improved.
Reviewer 3 Report
Comments and Suggestions for Authors
The authors explored the impact of micro- and nano-plastics (MNPs) on miRNAs, lncRNAs, and circRNAs, highlighting their ability to infiltrate organisms and accumulate, posing potential health risks. MNPs alter RNA expression, leading to immune, neurological, and other toxic effects. The study provides insights into MNP-induced biotoxicity. The conclusion is undoubtedly valid, and the manuscript clearly points out the data gaps that should be filled. However, some minor concerns should be addressed before considering for publication.
• Line 36: "Furthermore, MNPs can infiltrate cells, perturb intracellular signaling pathways and disturb immune homeostasis, ultimately triggering immunotoxicity in marine organisms and mammals". What is meant by "perturb intracellular signalling pathways"? Could you provide specific examples of these pathways and how they are affected by MNPs?
• More information about how exactly is mitochondrial dysfunction related to prolonged exposure to MNPs could be provided.
• Could authors elaborate on what is meant by "miRNAs exhibit tight temporal and spatial regulation within the nervous system"? How does this regulation affect their function in response to microplastic exposure?
• More details on how miRNAs serve as biomarkers for immunotoxicity caused by microplastic exposure could be added. What are the specific changes in miRNA levels that correlate with immune dysregulation?
• Can authors explain more about the NDK1-KSR1/2 cascade signaling pathway mediated by miR-38 and its role in transgenerational toxicity induced by PS-NPs?
• Besides NEAT1 and lncRNA-H19, are there other lncRNAs identified as part of ceRNETs with miRNAs under MNP exposure? What are their specific roles in cellular responses and toxicity?
• Are there studies that investigate the long-term effects of altered miRNA expression on chronic inflammatory conditions induced by MNPs?
• Could authors explain the mechanisms by which miR-21 overexpression in lupus nephritis contributes to the pathophysiology of SLE?
• Congratulations on using very current references in this work. However, it would be beneficial to consider updating references older than ten years, such as those from 2011, 2008, 2007, 2009, 2012, 2013, and 2014, if possible.
• Additionally, congratulations on the visually appealing and high-resolution figures included in this work.
